# Changes in access to viral load testing, incidence rates of viral load suppression and rebound following the introduction of the 'universal test and treat' guidelines in Cameroon: A retrospective follow-up analysis

C. E. Bekolo[1,2]*, S. A. Ndeso[2], L. L. Moifo[1], N. Mangala[3], J. Ateudjieu[1], C. Kouanfack[1], A. Dzudie[4], F. Thienemann[5], N. Tendongfor[2], D. S. Nsagha[2], S. P. Choukem[6]

1 Department of Public Health, Faculty of Medicine and Pharmaceutical Sciences, University of Dschang, Dschang, Cameroon, 2 Department of Public Health and Hygiene, Faculty of Health Sciences, University of Buea, Buea, Cameroon, 3 Department of Gynaecology and Obstetrics, University of Douala, Douala, Cameroon, 4 Department of Internal Medicine and Physiology, Faculty of Medicine and Biomedical Sciences, University of Yaoundé 1, Yaoundé, Cameroon, 5 Wellcome Centre for Infectious Diseases Research in Africa, Institute of Infectious Disease and Molecular Medicine, Faculty of Health Sciences, University of Cape Town, Cape Town, South Africa, 6 Department of Internal Medicine and Specialities, Faculty of Medicine and Pharmaceutical Sciences, University of Dschang, Dschang, Cameroon

* cavin.bekolo@univ-dschang.org

**Data Availability Statement:** Dataset has been uploaded as a supporting document.

## Abstract

Cameroon adopted and started implementing in 2016, the 'universal test and treat' (UTT) guidelines to fast-track progress towards the 95-95-95 ambitious targets to end the HIV epidemic. Achieving the third 95 (viral load suppression) is the most desirable target in HIV care. We aimed to evaluate the effectiveness of this novel approach on access to viral load testing (VLT), viral suppression (VLS), and viral load rebound (VLR). A retrospective cohort study was conducted at The Nkongsamba Regional Hospital to compare VLT outcomes between the pre-UTT (2002 to 2015) and the post-UTT (2016 to 2020) periods. We used a data extraction form to collect routine data on adult patients living with HIV. We measured uptake levels of the first and serial VLT and compared the incidence rates of VLS (VL<1000 copies/ml) and viral load rebound (VLR) before and after introducing the UTT approach using Kaplan Meier plots and log-rank tests. Cox regression was used to screen for factors independently associated with VLS and VLR events between the guideline periods. Access to initial VLT increased significantly from 6.11% to 25.56% at 6 months and from 12.00% to 73.75% at 12 months before and after introducing the UTT guidelines respectively. After a total observation time at risk of 17001.63 person-months, the UTT group achieved an incidence rate of 90.36 VLS per 1000 person-months, four-fold higher than the 21.71 VLS per 1000 person-months observed in the pre-UTT group (p<0.0001). After adjusting for confounding, the VLS rate was about 6-fold higher in the UTT group than in the pre-UTT group (adjusted Hazard Rate (aHR) = 5.81 (95% confidence interval (95%CI): 4.43–7.60). The incidence of VLR increased from 12.60 (95%CI: 9.50–16.72) to 19.11 (95%CI: 14.22–25.67) per 1000 person-months before and after the introduction of UTT guidelines

**Funding:** The authors received no specific funding for this work.

**Competing interests:** The authors have declared that no competing interests exist.

respectively. After adjusting, VLR was more than twice as high in the UTT group than in the pre-UTT group (aHR = 2.32, 95%CI: 1.30–4.13). Increased access to initial VLT and higher rates of VLS have been observed but there are concerns that the suppressed viral load may not be durable since the introduction of the UTT policy in this setting.

## Introduction

In 2016, Cameroon with an HIV (Human Immunodeficiency Virus) prevalence of 2.7% adopted and started implementing the 'universal test and treat' (UTT) policy to maximise the proportion of people living with HIV (PLHIV) on antiretroviral therapy (ART) and virally suppressed, that is, to fast-track progress towards the 95-95-95 global targets [1, 2]. Achieving the third 95 (viral load suppression—VLS) is the ultimate and most desirable target in the care of PLHIV and this can only be attained by an effective ART [3]. Under the UTT approach, ART initiation immediately after HIV diagnosis is recommended regardless of the CD4 count following the 2016 World Health Organization (WHO) consolidated guidelines for the treatment and prevention of HIV infection [3]. This UTT policy was informed by findings from clinical trials that demonstrated the potential of UTT to reduce HIV infection transmission and related morbidity and mortality through viral load suppression (VLS). Suppression of viral loads at 12 months was influenced positively across these randomised trials with rapid initiation [4–8]. However, there were concerns that adherence to ART and VLS amongst clinically healthy rapid ART initiators might be poor. However, these concerns were dispelled by findings from the HPTN 071 (PopART) community-randomised trial in Zambia and South Africa that showed that UTT did not affect adherence or VLS after three years of implementation [9]. Higher rates of VLS have been reported in many other settings at individual and population levels [10–14]. Yet, other settings have reported either a negative effect or no significant effect of UTT on VLS [15–17]. The observed inconsistency has been linked to methodological, social, political, economic, historical and health systems contextual factors that shape whether and how UTT is implemented or evaluated [5].

Viral suppression defined as a viral load<1000 copies/ml according to the WHO, can only be recognised when clients can assess viral load testing (VLT). VLT gives clients a measure of understanding, control, and motivation to adhere to treatment and understand their HIV infection [18]. Viral load testing has been recommended by WHO as the preferred monitoring approach to diagnose and confirm treatment failure [4, 19, 20]. However, VLT has not been feasible during the first decade of ART scale-up in many resource-constraint settings because it is expensive and technically complex [21, 22]. Only 24% of clients were accessing VLT in some clinics in Cameroon before the rollout of the UTT guidelines [23]. Under the UTT policy and thanks to support from the U.S. President's Emergency Plan for AIDS Relief (PEPFAR), in collaboration with the Global Fund to Fight AIDS, Tuberculosis, and Malaria, and the Governments of affected countries, access to free VLT has been expanding. The number of people accessing ART and thus requiring VLT under UTT has also been expanding. Implementing the Test and Treat policy in Cameroon, has been translated quantitatively into a significant increase in the number of persons accessing ART from 168431 (27.1% ART coverage) in 2015 to 350818 (70.6% ART coverage) in 2020 [24]. While this number is likely to increase rapidly, there are concerns over the country's ability to match this number with the increasing demand for VLT. Uncertainties also surround the ability of the country to keep the growing numbers of clients on ART until they achieve and sustain VLS. Monitoring expanded and timely access

to VLT and examining whether this has improved evidence-based clinical management, including sustained VLS or viral load rebound (VLR), are important goals of policy implementation and evaluation in Cameroon. We thus aimed to assess the changes observed in access to VLT, VLS and VLR rates since the introduction of the UTT guidelines in Cameroon.

## Methods

### Setting

The study was conducted at the Regional Hospital of Nkongsamba in the Mungo Division of the Littoral Region of Cameroon. It is a second-level reference public health facility with a catchment area of over 321,295 inhabitants [25]. Users of the clinic come from the city of Nkongsamba obviously but massively constitute referrals from the neighbourhood districts. The clinic was established in 2005 and offers voluntary HIV counselling and testing (VCT), ART and limited community outreach services to over 2000 patients on ART. HIV services are provided by multidisciplinary teams composed of physicians, health officers, nurses, pharmacy attendants, laboratory technicians, psychosocial/adherence supporters, and data personnel. As of 2016, per the national ART guideline, PLHIV are immediately linked to an ART clinic for a confirmatory test, counselling, adherence preparation and rapid ART initiation—including same-day ART for persons who are ready to start ART at the first clinical visit in the absence of CD4 testing that would otherwise delay ART initiation. Specimens collected for viral load testing were transported to The DREAM CENTRE of The St. Vincent de Paul Hospital in Dschang which hosts one of the few approved viral load platforms in the country. This centre was selected as a secondary site to ensure the completeness of data.

**Description and definition of Viral load monitoring indicators.** According to the national ART guidelines, HIV viral load testing is recommended six months and 12 months after initiating ART and every 12 months thereafter Viral load coverage was defined by the proportion of patients accessing a routine VL testing performed at the time it was due within the study period. ART patients are recommended to continue a first-line regimen with an undetectable viral load (VL<50 copies/ml) or a suppressed VL result defined as VL<1000 copies/ml. If they receive an initial unsuppressed VL result (> 1000 copies, patients are expected to receive enhanced adherence counselling with psychosocial workers and then to receive a repeat VLT three to 6 months later. If the result confirms virological failure (based on two consecutive viral load measurements above 1000 copies/mL three months apart, with adherence support following the first viral load test), they are eligible for switching to second-line regimens. Viral load rebound (VLR) was considered when a person on antiretroviral therapy (ART) has a detectable level of HIV in the blood after a period of undetectable levels.

### Study design

We conducted a facility-based retrospective cohort study using routine clinical service delivery accessed from medical records on the 23rd of October 2022. All PLHIV aged 15 and above enrolled into HIV care between 2002 and 2020 were eligible for inclusion. From the chart review, PLHIV enrolled before the year 2016 were included in the pre-UTT (unexposed or control) group while those enrolled from the year 2016 were included in the UTT (exposed) group. Based on days from HIV+ diagnosis to ART initiation, PLHIV were further stratified into the same-day initiation (SDI) group, rapid initiation group (1–7 days) or the deferred initiation group (8+ days). The groups were then followed up retrospectively until their first and repeat viral load measurements were done to compare their median time to VLT, and their rates of VLS and VLR (Fig 1).

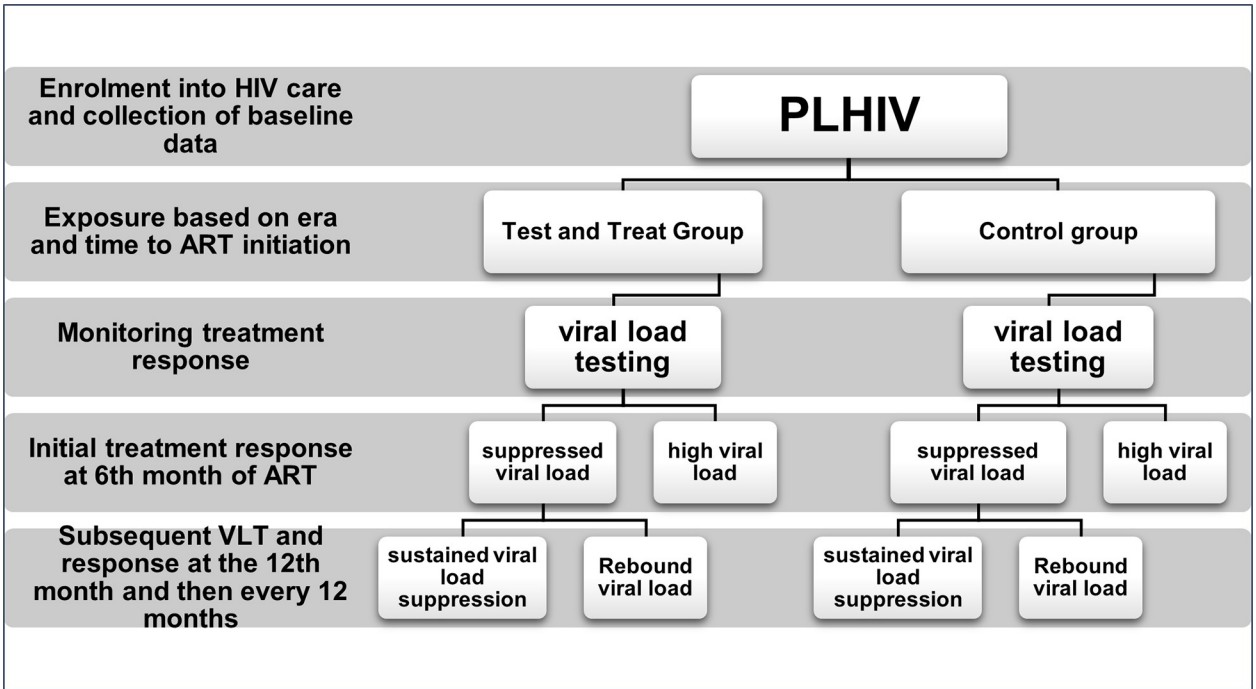

**Fig 1. Study cohort flow chart.**

## Data collection and study variables

Data were collected from individual patient medical records designed by the National AIDS Control Committee (NACC) for the standardised collection and reporting of data. We extracted at baseline the following putative exposure study variables using a data extraction form: socio-demographic characteristics: date of birth, gender, place of residence, occupation, alcohol and tobacco consumption, and matrimonial status; clinical features including date of HIV diagnosis, and WHO clinical stage at presentation, baseline CD4 count; treatment-related variables including date of ART initiation, ART regimen, adherence, cotrimoxazole preventive treatment (CPT); and the dates and titres of serial VLT.

## Ethics approval and consent to participate

Ethical approval was obtained from The Littoral Regional Ethics Committee for Research in Humans in Cameroon N˚ 2022/002/CE/CRERSH-LITTORAL. Permission to use data was duly obtained from the hospital management board. Consent from individual patients was not sought because we used routine data. However, all patient information was anonymised and de-identified before analysis. Access to the database was protected by a password.

## Data analysis

Data collected were exported to Stata 15.1 (StataCorp LLC, Texas 77845, USA) for statistical analysis. Summary statistics were presented as proportions for categorical variables and as means (standard deviations) for normally distributed continuous variables or medians (IQR-Interquartile Range) for skewed distributed continuous variables. The Chi-squared and the Fisher exact tests were used to compare proportions where appropriate.

Boxplots were used to graph the time to the initial and subsequent VLT between the UTT and standard groups and these times were compared using the Manova test.

We plotted and compared Kaplan Meier curves of months to VLS or VLR among the groups using the log-rank test. The follow-up period is the time between the enrolment into ART care and when the event of interest occurred (either VLT, VLS or VLR). The Cox regression model was used to screen for factors independently associated with VLS or VLR. These factors were selected based on a previous study in Kenya by Kimanga et al. [16]. Hazard ratios with their corresponding 95% confidence intervals (95%CI) and p-values were obtained from the final multivariable Cox model. Schoenfeld residuals were used for checking and testing the proportional hazards assumption.

## Results

### Baseline characteristics of participants

A total of 1651 persons diagnosed HIV positive between 2002 and 2020 were identified, of whom 1627 with identifiable data were included for analysis. Of those included, 756 (46.47%) were enrolled during the era of UTT with 545 (33.54%) initiated on ART on the same day of HIV diagnosis (Table 1). Participants enrolled under the UTT era were more likely to start ART on the same day of their HIV diagnosis (67.24%), to be married (35.16%), be living close to the HIV clinic (50.07%) or currently be in employment (68.98%). Baseline CD4 count testing was practically phased out during the UTT era (0.93%) and where available, it was targeted to those who were likely to have a titre below 350 cells/mm$^3$ but late clinical presentation was uncommon under UTT (17.56%).

### Universal 'test and treat' and access to viral load testing

The overall initial viral load testing (VLT1) coverage was 47.57% (95%CI: 45.14–50.02) and was similar between the pre-UTT and the UTT groups (47.53% vs. 47.62%) during the study period. However, the UTT group was accessing VLT1 at a rate of 90 VLT per 1000 person-months higher than the 22 VLT per 1000 person-months observed in the pre-UTT group (p<0.001). Access to VLT1 increased from 6.11% to 25.56% at 6 months and from 12.00% to 73.75% at 12 months before and after the introduction of UTT guidelines respectively (Fig 2A). Similarly, same-day initiators were accessing VLT1 earlier and faster than rapid and late initiators (Fig 2B). The median (IQR) time to serial VLT were significantly reduced after the implementation of UTT guidelines from 34.57(18.37–46.63) to 6.83(5.83–10.80) months for VLT1, 53.42(44.60–62.48) to 17.77(12.60–26.57) months for VLT2, 63.17(55.00–72.60) to 30.27(25.13–37.33) months for VLT3, and from 65.13(59.90–78.03) to 37.53(35.53–39.97) months for VLT4 (Manova, p<0.0001) (Fig 2C). A similar trend was observed with same-day initiators who had the shortest time to access repeat VLT compared to rapid and late ART initiators (Fig 2D).

### Universal 'test and treat' and changes in viral load suppression rates

Overall, the proportion of the cohort who achieved VLS was 78.60% (95%CI:75.56–81.42) but the UTT group had a higher proportion of VLS than the pre-UTT group (85.83% vs. 71.98%, p<0.001). The incidence rate of VLS was 35.35 cases per 1000 person-months generally after a total observation time at risk of 17001.63 person-months, with the UTT group attaining an incidence rate of 90.36 VLS per 1000 person-months higher than the 21.71 VLS per 1000 person-months observed in the pre-UTT group (p<0.0001) [Fig 3A]. Similarly, the proportion of VLS among same-day initiators was 87.50%, 80.95% among rapid initiators and 72.57%

**Table 1. Baseline characteristics of the study population by the period of ART initiation.**

| Variables | Before UTT n (%) | Under UTT n (%) | p-value |
|---|---|---|---|
| **Time to ART initiation** | | | |
| Same-day initiation (SDI) | 38 (4.36) | 507 (67.24) | |
| Rapid initiation (RI) | 135 (15.50) | 142 (18.83) | |
| Deferred initiation (DI) | 698 (80.14) | 105 (13.93) | |
| Total | 871 (100) | 754 (100) | <0.001 |
| **Age group (years)** | | | |
| < = 40 | 419 (48.44) | 350 (46.42) | |
| >40 | 446 (51.56) | 404 (53.58) | |
| Total | 865 (100) | 754 (100) | 0.417 |
| **Sex** | | | |
| Female | 576 (66.63) | 527 (70.36) | |
| Male | 287 (33.37) | 222 (29.64) | |
| Total | 860 (100) | 749 (100) | 0.108 |
| **Marital status** | | | |
| Single | 388 (46.92) | 344 (47.26) | |
| Married | 245 (29.63) | 256 (35.16) | |
| Divorced | 51 (6.17) | 35 (4.81) | |
| Widowed | 143 (17.29) | 93 (12.77) | |
| Total | 827 (100) | 728 (100) | 0.017 |
| **Area of residence** | | | |
| Within (Nkongsamba) Health District | 306 (35.75) | 374 (50.07) | |
| Out of (Nkongsamba) Health District | 463 (54.09) | 297 (39.76) | |
| Out of the (Littoral) Region | 87 (10.16) | 76 (10.17) | |
| Total | 856 (100) | 747 (100) | <0.001 |
| **Occupation** | | | |
| Unemployed | 318 (36.98) | 215 (29.78) | |
| Currently employed | 536 (62.32) | 498 (68.97) | |
| Retired | 6 (0.70) | 9 (1.25) | |
| Total | 860 (100) | 722 (100) | 0.007 |
| **Baseline CD4 testing** | | | |
| No | 46 (5.31) | 749 (99.07) | |
| Yes | 821 (94.69) | 7 (0.93) | |
| Total | 867 (100) | 756 (100) | <0.001 |
| CD4 count per mm$^3$ | | | |
| <350 | 442 (53.90) | 5 (83.33) | |
| > = 350 | 378 (46.10) | 1 (16.67) | |
| Total | 820 (100) | 6 (100) | 0.227 |
| **WHO Clinical Stage** | | | |
| Early (stages I & II) | 335 (48.83) | 582 (82.44) | |
| Advanced (stages III & IV) | 351 (51.17) | 124 (17.56) | |
| Total | 686 (100) | 706 (100) | <0.001 |
| **Smoking** | | | |
| No | 768 (88.17) | 676 (89.42) | |
| Yes | 103 (11.83) | 80 (10.58) | |
| Total | 871 (100) | 756 (100) | 0.429 |
| **Alcohol intake** | | | |
| No | 415 (47.65) | 311 (41.14) | |

*(Continued)*

**Table 1.** (Continued)

| Variables | Before UTT n (%) | Under UTT n (%) | p-value |
|---|---|---|---|
| Yes | 456 (52.35) | 445 (58.86) | |
| Total | 871 (100) | 756 (100) | 0.008 |

among deferred initiators corresponding to VLS rates of 77.88 vs. 38.69 vs. 25.15 VLS per 1000 person-months respectively (p<0.001) [Fig 3B].

**Predictors of viral load suppression.** After adjusting for confounding, there was still very strong evidence that the VLS rate was approximately 6-fold higher in the UTT group than in the pre-UTT group (adjusted Hazard Ratio [aHR] = 5.81 (95%CI: 4.43–7.60, p<0.001)] but there was little, or no evidence left to support the effect of same-day initiation on VLS. Other factors significantly associated with higher VLS rates were age above 40 years [aHR = 1.22 (95%CI: 1.01–1.47, p = 0.033)], and early clinical presentation [aHR = 1.42 (95%CI: 1.15–1.76, p = 0.001)]. (Table 2).

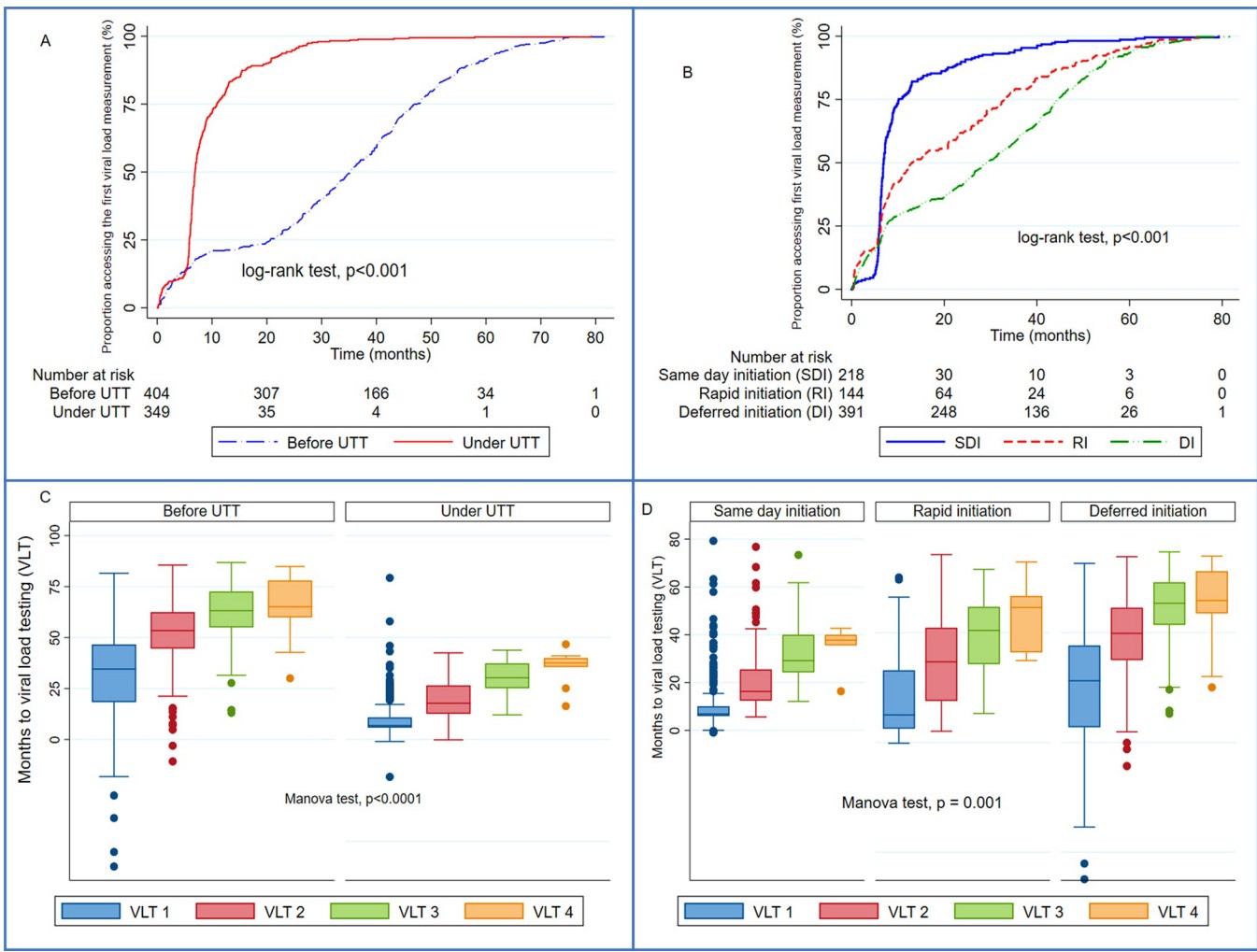

**Fig 2. Cumulative access to initial viral load (A and B) and time to repeat viral (C and D) measurements across ART initiation guidelines.**

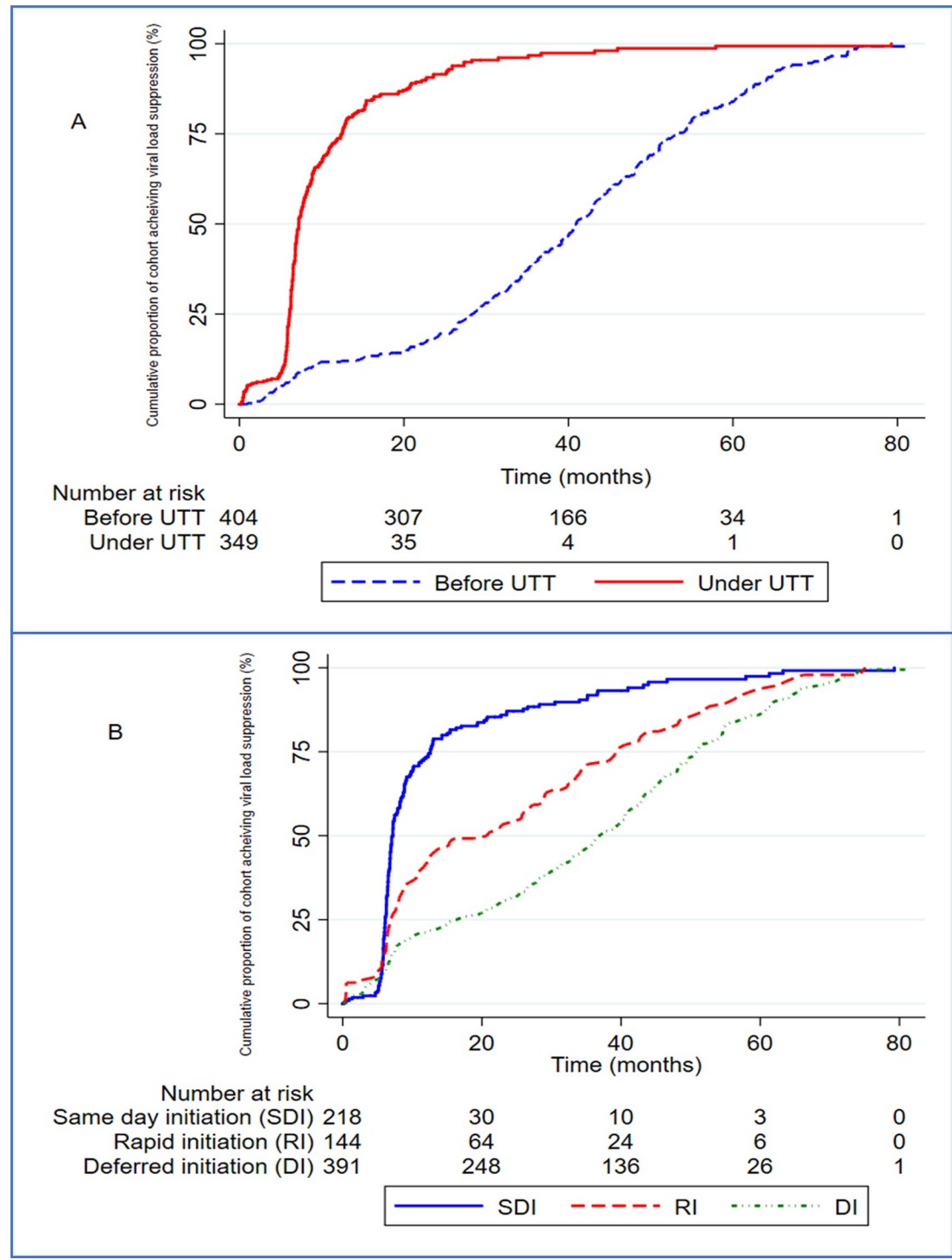

**Fig 3. Cumulative rates of viral load suppression across ART initiation periods and strategies.**

**Table 2. Factors independently associated with viral load suppression.**

| Factors* | Rate of VL suppression (per 1000 person-months) | HR (95%CI) | p-value | aHR (95%CI) | p-value |
|---|---|---|---|---|---|
| **ART initiation strategy** | | | | | |
| Same day | 77.88 | 3.39 (2.79–4.13) | <**0.001** | 1.25 (0.96–1.63) | **0.097** |
| Rapid | 38.69 | 1.56 (1.26–1.94) | <**0.001** | 1.14 (0.89–1.47) | **0.287** |
| Deferred | 25.15 | 1 | | 1 | |
| **Cohort** | | | | | |
| Before UTT | 21.72 | 1 | | 1 | |
| Under UTT | 90.36 | 6.65 (5.45–8.10) | <**0.001** | 5.81 (4.43–7.60) | <**0.001** |
| **Age (years)** | | | | | |
| < = 40 | 33.16 | 1 | | 1 | |
| >40 | 37.17 | 1.15 (0.97–1.35) | **0.098** | 1.22 (1.01–1.47) | **0.033** |
| **Baseline CD4 testing** | | | | | |
| No | 85.01 | 1 | | 1 | |
| Yes | 21.42 | 0.17 (0.14–0.21) | <**0.001** | 0.60 (0.28–1.30) | 0.196 |
| **WHO Clinical Stage** | | | | | |
| Early (stages I & II) | 41.87 | 1 | | 1 | |
| Advanced (stages III & IV) | 29.01 | 0.68 (0.56–0.83) | <**0.001** | 0.70 (0.57–0.87) | **0.001** |
| **Residence** | | | | | |
| Within (Nkongsamba) Health District | 38.74 | 1 | | 1 | |
| Out of (Nkongsamba) Health District | 32.05 | 0.84 (0.71–0.99) | **0.046** | 1.03 (0.85–1.25) | 0.739 |
| Out of the (Littoral) Region | 37.30 | 0.92 (0.67–1.26) | 0.600 | 1.21 (0.85–1.71) | 0.282 |
| **Occupation** | | | | | |
| Unemployed | 34.29 | 1 | | 1 | |
| Currently employed | 34.67 | 0.98 (0.82–1.16) | 0.818 | 0.82 (0.67–0.99) | **0.041** |
| Retired | 101.94 | 2.96 (1.39–6.31) | **0.005** | 1.62 (0.71–3.71) | 0.254 |

* Controlled for age, sex, area of residence, occupation, clinical presentation, CD4 count, period and type of treatment initiation

## Universal 'test and treat' and changes in viral load rebound rates

The overall risk of viral rebound was 23.31% (95%CI: 19.24–27.77), with no significant differences across the treatment guidelines periods. The group-specific risks were 23.66% in the UTT group, 23.19% in the pre-UTT group, 24.07% in the same-day initiation group, 14.74% in the rapid initiation group and 27.37% in the deferred initiation group. After a total period of observation of 6235.10 person-months, the overall VLR rate was 14.91 per 1000 person-months increasing from 12.60 (95%CI: 9.50–16.72) to 19.11 (95%CI: 14.22–25.67) per 1000 person-months before and after the introduction of UTT guidelines respectively. The median months of a durably low viral load reduced from 38.97 (IQR: 19.03–53.73) before UTT to 24.10 (IQR: 17.30–43.47) under UTT(Fig 4A). The rates of VLR were 20.28 (95%CI: 13.81–29.79) in the same-day initiation group, 9.64 (95%CI: 5.71–16.28) in the rapid initiation group and 15.39 (95%CI: 11.73–20.20) in the deferred initiation group. The median months of a durably low viral load were respectively 27.77 (IQR: 15.17–43.47), 38.97 (IQR: 24.10 -.) and 34.47 (IQR: 17.70–52.93) in the same-day initiation, rapid initiation, and deferred initiation groups (Fig 4B).

**Predictors of viral load rebound.** After adjusting for confounding, the incidence rate of VLR was twice as high in the UTT group than in the pre-UTT group [adjusted Hazard ratio (aHR) = 2.32, 95%CI: 1.30–4.13, p = 0.004]. There was no significant effect of same-day initiation on VLR, but rapid initiation was associated with a 44% reduction in VLR compared to the

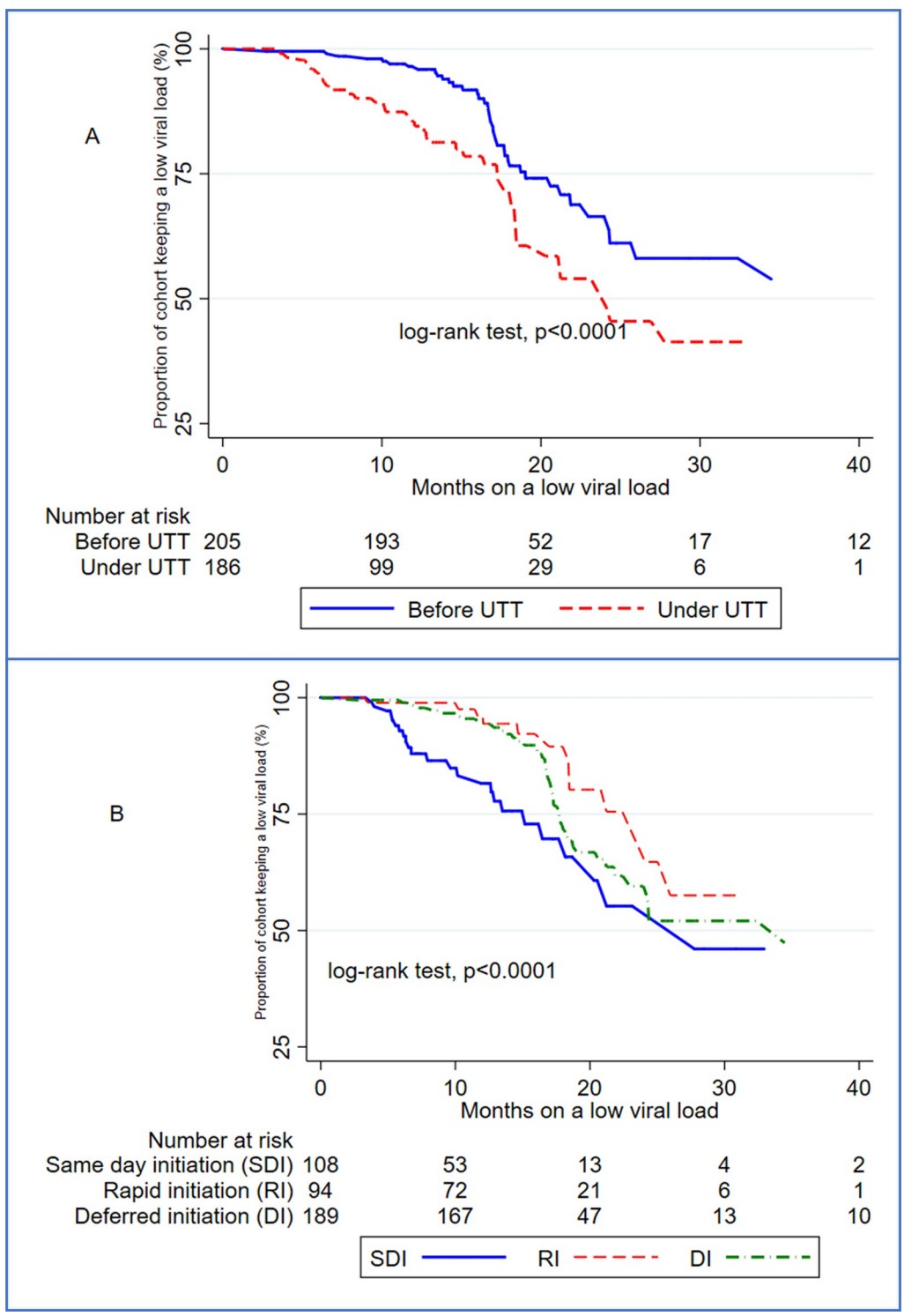

**Fig 4. Retention on low viral load across ART initiation guidelines and periods.**

**Table 3. Factors independently associated with viral load rebound.**

| Factors* | Rate of VL rebound. (per 1000 person-months) | HR (95%CI) | p-value | aHR (95%CI) | p-value |
|---|---|---|---|---|---|
| **ART initiation strategy** | | | | | |
| Same day | 20.28 | 1.75 (1.09–2.82) | **0.021** | 0.94 (0.50–1.77) | **0.860** |
| Rapid | 9.64 | 0.66 (0.37–1.20) | **0.178** | 0.56 (0.28–1.11) | **0.097** |
| Deferred | 15.39 | 1 | | 1 | |
| **Cohort** | | | | | |
| Before UTT | 12.60 | 1 | | 1 | |
| Under UTT | 19.10 | 1.99 (1.29–3.06) | **0.002** | 2.32 (1.30–4.13) | **0.004** |
| **Sex** | | | | | |
| Female | 13.12 | 1 | | 1 | |
| Male | 19.11 | 1.40 (0.92–2.13) | **0.111** | 1.69 (1.04–2.75 | **0.034** |
| **Smoking** | | | | | |
| No | 13.61 | 1 | | 1 | |
| Yes | 24.79 | 2.35 (1.39–3.96) | **0.001** | 2.29 (1.24–4.23) | **0.008** |
| **Adherence** | | | | | |
| Poor (<95%) | 19.05 | 1 | | 1 | |
| Good (≥95%) | 12.90 | 0.60 (0.38–0.94) | 0.026 | 0.58 (0.36–0.94) | **0.028** |
| **Clinical presentation** | | | | | |
| Early (Stages I & II) | 14.09 | 1 | | 1 | |
| Advanced (Stages III & IV) | 19.65 | 1.37 (0.87–2.17) | **0.175** | 1.79 (1.04–3.06 | **0.036** |

* Controlled for age, sex, area of residence, occupation, lifestyle, clinical presentation, CD4 count, period and type of treatment initiation

deferred initiation group. Other factors associated with a high VLR were poor adherence to ART [aHR = 1.72 (95%CI: 1.06–2.81, p = 0.028)], current smoking status [aHR = 2.29 (95%CI: 1.24–4.23, p = 0.008)], male sex [aHR = 1.69 (95%CI: 1.04–2.75, p = 0.034)], and clinical presentation with advanced HIV infection [aHR = 1.79 (95%CI: 1.04–3.06, p = 0.036)] (Table 3).

## Discussion

In this institutional-based study, we sought to assess the effect of the UTT guidelines on viral load coverage and the incidence rates of viral load suppression and rebound in Cameroon. Firstly, we found that though viral load coverage was persistently low as less than half of eligible clients could access their initial VLT, the delay to the initial VLT was significantly reduced by one quarter since introducing the UTT guidelines in 2016. Secondly, though a VLS of 86% was still below the 95% target, it had increased by 14% under UTT and clients in the UTT era were achieving VLS at a rate six-fold higher than those in the pre-UTT era. Thirdly, the overall risk of VLR was moderately high, with an incidence rate more than twice as high in the UTT era than in the pre-UTT era.

Timely access to VLT significantly improved under the UTT era as the country shifted from targeted to routine VLT as per the 2013 WHO guidelines [2, 19]. This strategic and programmatic transition was made feasible because the number of conventional VLT platforms and point-of-care (POC) machines had quadrupled since the implementation of UTT in the country [24]. In addition, there was the removal of user fees, continuing mentorship of healthcare workers, the establishment of hub-and-spoke laboratories to reorganise the transport system, mobile coolers to strengthen the cold-chain, use of solar panel installations and refrigerators, as well as the use of psychosocial and community health workers to remind clients to visit the clinic for VLT [24, 26]. Despite the scaling up of VLT, overall viral load testing

coverage in this study and the country at large (47.6% vs.58.1%) remains low as significant challenges persist including prolonged and frequent commodity stock-outs, limited cold-chain capacity, long distances to health facilities and poor roads [21, 23, 27]. The persistently low VLT coverage is linked to the increasing number of patients who are also accessing ART because of UTT and who eventually require a VLT. Though the supply of VLT platforms has equally increased, it has not been able to match the exponential increase in demand for VLT under the UTT era. Our study site still serves as a spoke laboratory and must send specimens to the hub for analysis. Expanding VLT platforms to reach most or all HIV clinics is mandatory to achieve optimal outcomes under the UTT policy.

After the introduction of the UTT guidelines, the overall proportion of VLS and the median time to viral suppression did improve considerably thus taking the cohort closer to the third 95 and, providing important and timely information to providers and clients on the effectiveness of antiretroviral therapy. Increasing access to ART which leads to the suppression of viral replication has been the main driver of VLS. Following the adoption of the UTT guidelines, ART coverage increased from 27% in 2015 to 71% in 2020 allowing many PLHIV to receive ART and thus achieve VLS [28]. Also, in this study, early presentation increased from 48.83% before UTT to 82.44% post-UTT, and there was strong evidence that early presentation was associated with a higher incidence of VLS. Early initiation of ART which is the hallmark guideline of the UTT policy, leads to the establishment of low HIV reservoir size and has been shown to benefit PLHIV by reducing plasma viral load, gut damage, microbial translocation and subsequent systemic immune activation [29]. Moreover, the availability and transition to optimised, potent, and well-tolerable ART regimens during the era of UTT must have had a significant impact on adherence and viremia. While the shift from targeted to routine VL monitoring was another major determinant of the changes observed in the rate of VLS, efforts to improve adherence and retention through differentiated care, task shifting, and decentralised care were equally upgraded during UTT. Despite these tangible improvements, VLS in this setting is yet to meet the 95% target but fits within the global range of 81– >98% [30]. UTT alone may not be sufficient to achieve universal VLS, other individual and health system factors should be considered including efforts to further improve early presentation, target young adults and fight against unemployment may also contribute to achieving the third 95 in the era of UTT.

While this study has indicated that there have been improvements in VLS under UTT, we have concerns about sustaining a durably low viral load under the UTT policy. We observed that under UTT the rate of viral rebound had doubled. This was an unexpected finding on the premises that initiation of ART early after HIV infection will substantially restrict the size of the HIV reservoir and thus the time to viral rebound [31, 32]. Early clinical presentation was indeed associated with a lower viral rebound rate as an independent factor alongside other factors like good adherence, female gender, and the non-smoking habit. A population-based survey in Uganda found that the prevalence of VL rebound declined from 4.4% to 2.7% after UTT [33]. Though there are methodological differences between our study and the latter, there are a few contextual and plausible reasons to explain our findings. First, less frequent follow-up clinical visits and multi-month drug dispensations are common under UTT, particularly among stable patients. This reduces effective contact between clients and their healthcare providers, potentially exposing both parties to a complacent relationship and the former to inadequate adherence support. Research into whether long appointment spacing affects low viral load durability is thus necessary. Also, routine, timely and repeated VLT were more common under the UTT policy meaning that there was a higher probability of detecting viral rebounds than ever before. A multicentric study will be necessary to provide further evidence of the effect of UTT on viral load rebound.

This study owes its strength to its potential to have measured both the cumulative incidence and incidence density of events necessary for monitoring both individual patient outcomes and programme performances. The relatively large sample size prompts generalisability. The contextual and secular changes, residual confounding and possible selection bias are limitations to this study. These limitations make it difficult to interpret the observed changes in viral load monitoring as being attributed to the effect of UTT per se. A potential solution to overcome these limitations is to conduct difference-in-differences (DiD), interrupted time series or regression discontinuity analyses to eliminate the effect of trends over time and permanent differences between the groups if assumptions are met. The quality of routine data must have improved between the two periods thus contributing to changes observed in VL monitoring.

## Conclusion

Overall VLT coverage has not improved significantly under UTT. However, delays to accessing initial and repeated VLTs have been reduced drastically and higher rates of VLS have been observed under the UTT. There are also concerns that the suppressed viral load may not be durable under UTT. Efforts to further improve access to viral load monitoring and sustain VLS are critical to achieving optimal outcomes under UTT.

## Supporting information

**S1 Data. Dataset of variables.**
(XLSX)

## Acknowledgments

We are grateful to the staff and patients of The Nkongsamba Regional Hospital & and The Centre Dream of St Vincent de Paul Hospital in Dschang, Cameroon for their collaboration in data collection.

## Author Contributions

**Conceptualization:** C. E. Bekolo, S. A. Ndeso, S. P. Choukem.

**Data curation:** C. E. Bekolo, L. L. Moifo.

**Formal analysis:** C. E. Bekolo.

**Investigation:** C. E. Bekolo, L. L. Moifo, N. Mangala.

**Methodology:** C. E. Bekolo, S. A. Ndeso, C. Kouanfack, A. Dzudie, F. Thienemann.

**Project administration:** N. Mangala, J. Ateudjieu, N. Tendongfor.

**Resources:** N. Mangala.

**Supervision:** S. A. Ndeso, J. Ateudjieu, C. Kouanfack, A. Dzudie, F. Thienemann, N. Tendongfor, D. S. Nsagha, S. P. Choukem.

**Validation:** S. A. Ndeso, A. Dzudie, N. Tendongfor, D. S. Nsagha, S. P. Choukem.

**Writing – original draft:** C. E. Bekolo.

**Writing – review & editing:** C. E. Bekolo, S. A. Ndeso, J. Ateudjieu, C. Kouanfack, A. Dzudie, F. Thienemann, N. Tendongfor, D. S. Nsagha, S. P. Choukem.

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
