## [Decision Letter · Decision Letter 0]

6 Feb 2024

PGPH-D-23-02088

Changes in access to viral load testing, incidence rates of viral load suppression and rebound following the introduction of the ‘Universal Test and Treat’ guidelines in Cameroon: a retrospective follow-up analysis.

Dear Dr. Bekolo,

Thank you for submitting your manuscript to PLOS Global Public Health. After careful consideration, we feel that it has merit but does not fully meet PLOS Global Public Health’s publication criteria as it currently stands. Therefore, we invite you to submit a revised version of the manuscript that addresses the points raised during the review process.

EDITOR'S COMMENT: In addition to the reviewers' comments, I have a comment for your response:

I noticed that you recently published a similar paper entitled: "The effect of the Universal Test and Treat policy uptake on CD4 count testing and incidence of opportunistic infections among people living with HIV infection in Cameroon: a retrospective analysis of routine data".  Please provide a sound scientific rationale for the submitted work and clearly reference and discuss the existing literature.

We look forward to receiving your revised manuscript.

Kind regards,

Omotayo Fatokun, PhD., MPharm., MPH

Guest Editor

Journal Requirements:

Additional Editor Comments (if provided):

Reviewers' comments:

Reviewer's Responses to Questions

**Comments to the Author**

1. Does this manuscript meet PLOS Global Public Health’s publication criteria? Is the manuscript technically sound, and do the data support the conclusions? The manuscript must describe methodologically and ethically rigorous research with conclusions that are appropriately drawn based on the data presented.

Reviewer #1: No

Reviewer #2: Partly

2. Has the statistical analysis been performed appropriately and rigorously?

Reviewer #1: No

Reviewer #2: No

3. Have the authors made all data underlying the findings in their manuscript fully available (please refer to the Data Availability Statement at the start of the manuscript PDF file)?

Reviewer #1: Yes

Reviewer #2: Yes

4. Is the manuscript presented in an intelligible fashion and written in standard English?

Reviewer #1: No

Reviewer #2: Yes

5. Review Comments to the Author

Reviewer #1: Review 1

Title: Changes in access to viral load testing, incidence rates of viral load suppression and rebound following the introduction of the ‘Universal Test and Treat’ guidelines in Cameroon: a retrospective follow-up analysis.

Abstract

Background: I guess retention also is a desirable outcome in HIV care. I am wondering why the authors did not look at retention as while. Why didn’t the authors also look at the clinical and demographic factors associated with VLS before and after UTT; informing cardinal for targeted intervention? If such information is available, it will be nice to see such an analysis.

Methods: authors can consider segregating the pre- and post-UTT period. E.g before (between 2002 and 2015) and after (between 2016 and 2020), you can also include the months. Was the data collected from both children and adults? Include the information on points through which data were collected e.g. 6, 12, 24 months etc and state the tool used to collect the data. The period before UTT was longer than the after. Why didn’t the authors have an equal follow-up period? What was the primary outcome? If viral suppression, how was it defined? I encourage the authors to include such information. In the last sentence, authors can specify the VL events (VLS and VLR). VLR should be stated in the aim.

Results: first paragraph, was there a statistical difference in the proportion of VLT? Does it mean data were only collected at two points before and after UTT i.e. 6 and 12 months? If yes, how was it done given that data was collected between 2002 and 2020? What were the inclusion criteria for participants in the pre-and post-UTT? I thought that each individual was followed for a specific period before and after UTT. OR did you follow up with everyone and UTT was part of the variables? If yes, what were the inclusion criteria as of 2002? What was the average overall follow-up period? And also segregate it for before and after UTT. CI should be written as “confidence interval (CI)” since it appears for the first time. Do the same for “aHR”. Say “The incidence of VLR was more than twice as high in the UTT group than in the pre-UTT group (aHR = 2.32, 95%CI: 1.30 – 4.13) after adjusting for …………………..”. I encourage authors to indicate the proportions of VLS. I did not see the results on VLR.

Conclusion: Where are the concerns coming from over the durability of VLS? What informs the concerns? I encourage the authors to highlight a recommendation/s based on their findings.

Introduction

- “Achieving the third 95 (viral load suppression - VLS) is the ultimate and most desirable target in the care of PLHIV and this can only be attained by an effective ART”, cite. And please note that retention is also cardinal in achieving viral suppression.

- 3 years should be written as “three years or three (3) years”

- “Viral suppression defined as a viral load<1000 copies/ml according to the WHO, can only be recognised or achieved when clients can assess viral load testing (VLT)”, I don’t think viral load testing can help to achieve VL <1000, I do agree on the recognition part. Probably retention is one of the factors imperative to achieving VLS. I propose you delete “or achieved” and information on characteristics important in achieving VLS.

- “VLT gives clients a measure of understanding, control, and motivation to adhere to treatment and understand their HIV infection”, back this information with scientific evidence. Note that other clients do not bother about VLT as long as they can get their meds. Based on my experience, vein puncture does not sit well with most ART clients.

- The gap is not really coming out; does it mean viral load coverage and VLS were unknown before and after UTT in Cameroon? See our publication where we also looked at VLS https://pubmed.ncbi.nlm.nih.gov/37900026/.

- The authors should do more literature review on VLT, VLS and VLR comparing the before and after UTT.

Methods

- I encourage the authors to have a “Study setting” separate from viral load monitoring. I think viral load monitoring can be under “Data collection and study procedure” OR it can have its section. Indicate the study design before the viral load monitoring BUT I don’t understand why viral load monitoring has such details. Did the authors use retrospective data? If yes, the sample size estimation, data abstraction process (sampling method/s), data quality control, definitions of outcomes, data analysis, and ethical consideration would be of great concern to me.

- If the authors conducted a prospective study in the period after the test and treat policy, the “Study procedure” section would address the issues of viral load monitoring. However, the abstract has already stated that it was a retrospective analysis. General guidance on the flow of methods section: “1) study setting 2) study design and population 3) sample size estimation, 4) data abstraction process (sampling method/s), 5) data quality control, 6) definitions of outcomes, 7) data analysis, and 8) ethical consideration”.

- The whole issue of viral load monitoring is confusing.

- The definitions for VLS, VLR and viral load coverage/testing can be defined under the section “Definitions of outcomes or operational definitions”

- Figure 1, I don’t seem to fit it anywhere in the document. Maybe authors can consider having a flowchart for inclusion criteria so it can show numbers analysed for both before and after UTT.

- “while those enrolled after the year 2006 were included in the UTT (exposed) group.”, I guess authors meant 2016 NOT 2006. Please correct accordingly. The periods before (2002 to <2016) and after (2016 to 2020) UTT are not similar. Is there a special reason for this?

- Authors should state whether they included clients who were just starting ART at baseline or not.

- Authors should state the follow-up periods e.g. at baseline, 6, 12, or 24 months

- “The groups were then followed up retrospectively until their first and repeat viral load measurements were done to compare their median time to VLT, and their rates of VLS and VLR.” Since authors were looking for this why didn’t they look at individuals initiating ART before and after UTT and follow them up for a period of one year for each period? I can’t understand why authors had to go back to 2002. If it is the issue of VLR they would have extended the period to 2 or 3 pre- and post-UTT.

- Data collection and study variables: authors should state what type of data were collected at what period e.g. baseline, 6 and 12 months.

- Authors should be detailed on how data were collected so we can connect it with the data analysis plan.

- “The data set was explored for logical inconsistencies, illegal codes, omissions and improbabilities by tabulating, summarising, describing, and plotting variables”, was this part of data cleaning processing? This is what is usually done in the background and rarely mentioned in the manuscript. You may need to remove this part.

- For missing values, authors may consider an imputation method.

- “Viral load coverage was defined by the proportion of patients accessing a routine VL testing performed at the time it was due within the study period”, should be part of the “Operational definition”.

- Say “Schoenfeld residuals was used for checking and testing the proportional-hazards assumption”

Results

- Table 1, first row should include the total number (%) for individuals before and after UTT.

- “545 (33.54%) initiated on ART on the same day of HIV diagnosis”, this information is not there in Table 1.

- “Participants enrolled under the UTT era were more likely to start ART on the same day of their HIV diagnosis (67.24%)”, this is not a surprising finding to due guideline issue as compared to the period before UTT. Hope the discussion have taken care of this.

- Table 1, please countercheck because some of the variable they don’t add upto 100% e.g marital status. Check all the variable and revise accordingly.

- Table 1, why was age categorized in that way? You may want to categorize age based on some similarity with regards to sociodemographic or other factors responsible for the outcome. I don’t think there are similarities in VLS between adolescent and/or young adults (15 to 19 or 19 to 25 years) and those aged 25 to 35 or those aged 36 to 40 years. Check some recommendations from WHO.

- Table 1, variable area of residence, it would be more understandable by other HIV program managers from other countries if it was categorized as “rural or urban”. In its current only Cameroon can benefit from the information.

- Did you have any missing values in some variables, if yes, it would be nice to indicate them.

- Table 1, CD4 should be categorized based on some source or clinical knowledge. Immunocompetence if am not mistaken >=500. If you want to leave the threshold of 350 define as such in operational definition section.

- Same of the initiators should be written in full to avoid confusing the readers e.g. SDI, RI and DI. Sometimes too many abbreviations are confusing.

- The narration for Table 2 should have own subheading. Do the same for Table 3.

- “Overall, the proportion of the cohort who achieved VLS was 78.60% (95%CI:75.56 –81.42) but the UTT group had a higher proportion of VLS than the pre-UTT group (85.83% vs. 71.98%, p<0.001).” which table shows this information? I suggest Table 2 and 3 should have results for a chi-square test.

- What was the primary outcome between VLS and VLR?

- First define aHR and thereafter use the initials

- Did you collect data on the type of ART?

- What do you mean by “time and type of treatment initiation”?

- Table 2 and 3, the variables used for controlling the model should not be in the table. You can have a supplementary table for them.

- What does “t” mean “of t SDI”?

- “Other factors significantly associated with higher VLS rates were age above 40 years, early clinical presentation, and current employment status”, narrate these results as opposed just stating like this.

- Table 2 has no lifestyle variables.

- Similarly, under this section “Universal ‘test and treat’ and changes in viral load rebound rates” some results ae not shown either in a table format or figure.

- Was the rate of VLR statistically different for pre- and post-UTT?

- In Table 3, all the significant variables should be narrated.

- I didn’t see alcohol consumption in the models. Maybe instead of having a supplementary table, show all the variables in your models.

- You can also consider looking at the factors associated with VLS and VLR for Pre- and post-UTT. It will be great to see the drivers of these outcomes in the period before and after the test and treat; tables can be supplementary.

- Include the year in your analysis if you have such information, to see the trends of VLS and VLR across the years

Discussion:

- No need to restate the aim of the study.

- How was timely viral load testing calculated when individuals have a specific period to come to the clinic based on the guideline? It would be nice to see a proportion of individuals who had a viral load test at specified time intervals in line with your national guidelines.

- The first paragraph should summarize your findings in line with your aim.

- “The persistently low VLT coverage is linked to the increasing number of patients who are also accessing ART because of UTT and who eventually require a VLT”, I don’t think this is a reason enough for low VL coverage. If you have the necessary and enough human resources, equipment, testing facilities and reagents, you can still have a large proportion of those who have a VL test.

- What do you mean by early presentation? Or which results represent early presentation?

- You can also discuss the significant findings on the factors associated with VLS and compare with previous literature like ours and many more.

- “While this study has indicated that there have been improvements in VLS under UTT, there is uncertainty as to whether UTT may potentially delay the time to viral rebound after initial viral suppression”, uncertainty by who?

- How many months is “multi-month drug dispensations”?

- Does it mean, the authors had limited literature to compare with theirs? Authors have more probable explanations compared to the cited previous literature. I encourage the others to do more literature reviews.

- Discuss the other variables significantly associated with VLR.

- In the section for study limitations and strengths, take the recommendation/s to the conclusion.

- Interrupted time series would also be a good quasi-experimental study design to generate good estimates for your counterfactuals.

- Why do you expect typing errors?

- Given that this was longitudinal data, did you have LTFU?

- The last sentence of the discussion section does not connect well and is kind of confusing. Please revise accordingly.

Conclusion

- This section should summarize your main finding, the implication/s of your findings and recommendation/s.

- Are the concerns part of your findings? If not please revise accordingly.

Acknowledgement

- Authors can specify why they are grateful to the patients and staff unlike leaving it hanging.

Reviewer #2: Thank you for submitting a manuscript titled,

“Changes in access to viral load testing, incidence rates of viral load suppression and rebound following the introduction of the ‘Universal Test and Treat’ guidelines in Cameroon: a retrospective follow-up analysis”

This is a very intriguing subject and adds a lot of knowledge and insights into the UTT approach to HIV elimination.

However, I have some concerns which need to be addressed before the article can be published.

1. Title

I think it would read better if the title was changed to “Trends in viral load access and testing, incidence rates of viral load suppression and rebound following the introduction of the ‘Universal Test and Treat’ guidelines in Cameroon: a retrospective follow-up analysis.”

This would inform the analysis approach and enable the researchers to clearly show Cthe yearly changes, rather than just categorizing as Pre-UTT and UTT.

2. Methodology.

You mentioned that the clinic where the study was conducted started in 2005, but your review included those who initiated ART from 2002, how accurate was this information? And how many had initiated ART between 2002-2005? It would have been better to exclude those who were started on ART before the clinic was opened or include as part of the limitation for the study.

3. Results analysis.

Its better to present a trends analysis from 2002 or 2005 to show how viral coverage and suppression has been changing over time. This will help to explain the micro changes which could have taken place before and after UTT, eg changes in ART initiation guidelines and viral load accessibility policies.

For Viral load suppression, table2, whereas statistically it is correct to say that “the hazard of viral load suppression was 6 times higher in UTT than in pre-UTT”, programmatically it is rather confusing. I therefore suggest that a reanalysis is done with the outcome of interest as viral load non-suppression. This will make it clearer to a programmer.

6. PLOS authors have the option to publish the peer review history of their article (what does this mean?). If published, this will include your full peer review and any attached files.

**Do you want your identity to be public for this peer review?** For information about this choice, including consent withdrawal, please see our Privacy Policy.

Reviewer #1: No

Reviewer #2: No

---

## [Editor Report · Decision Letter 1]

29 Feb 2024

Changes in access to viral load testing, incidence rates of viral load suppression and rebound following the introduction of the ‘Universal Test and Treat’ guidelines in Cameroon: a retrospective follow-up analysis.

PGPH-D-23-02088R1

Dear Dr. Bekolo,

We are pleased to inform you that your manuscript 'Changes in access to viral load testing, incidence rates of viral load suppression and rebound following the introduction of the ‘Universal Test and Treat’ guidelines in Cameroon: a retrospective follow-up analysis.' has been provisionally accepted for publication in PLOS Global Public Health.

Best regards,

Omotayo Fatokun, PhD., MPharm., MPH

Guest Editor